# Variation and Abundance of Resistant Starch in Selected Banana Cultivars in Uganda

**DOI:** 10.3390/foods13182998

**Published:** 2024-09-21

**Authors:** Ali Kajubi, Rhona Baingana, Moses Matovu, Ronald Katwaza, Jerome Kubiriba, Priver Namanya

**Affiliations:** 1National Agricultural Research Laboratories (NARL), Kampala P.O. Box 7065, Uganda; mousa2k@yahoo.com (M.M.); rnldkatwaza@gmail.com (R.K.); jkubiriba2012@gmail.com (J.K.); bwesigyep@gmail.com (P.N.); 2College of Natural Sciences, Makerere University, Kampala P.O. Box 7062, Uganda; rhona.baingana@mak.ac.ug

**Keywords:** resistant starch, banana, nutraceutical, chemometrics, crystalline, microarchitecture

## Abstract

The physiochemical, structural, and molecular characteristics of starch influence its functional properties, thereby dictating its utilization. The study aimed to profile the properties and quantity of resistant starch (RS) from 15 different banana varieties, extracted using a combination of alkaline and enzyme treatments. Granular structure and molecular organization were analyzed using light microscopy, scanning electron microscopy (SEM), and Fourier transform infrared spectroscopy (FTIR). The physiochemical and functional properties were also investigated. RS content ranged from 49% to 80% without significant relationship to amylose (AM) (r = −0.1062). SEM revealed significant microarchitectural differences on the granules potentially affecting granule digestibility. FTIR and chemometrics identified differences in the crystalline peaks, yielding varying degrees of the molecular order of the RS polymers that aid in differentiating the RS sources. Despite similar solubility and swelling profiles, the pasting profiles varied across varieties, indicating high paste stability in hydrothermal processing. Clarity ranged from 43% to 93%, attributed to amylose leaching. This study highlights that RS from bananas varies in quantity, structure, and functionality, necessitating individualized approaches for processing and utilization.

## 1. Introduction

Bananas (*Musa* sp.), which are one of the most economically important food crops after rice, wheat, and maize [1], accumulate large amounts of starch, ranging from 80–90% (dwb) [2]. Many of the domesticated banana cultivars are natural mutants with a triploid genome such as dessert apple banana (AAB), cooking banana (AAA), and plantain (AAB or ABB) [3]. East African highland banana cultivars (EAHB) (*Musa AAA-EA*) comprise cooking bananas, which form over 80% of the cultivars in the Great Lakes Region and are a source of food and income for over 40 million people in the East African Community (EAC) [4].

Starch, which is the most important source of carbohydrate in human nutrition, is made up of linear and branched polymers of D-glucose molecules linked via glycosidic bonds. The relative proportion of these polymers and their organizational structure within starch granules dictate the functional properties of starch and the corresponding range of industrial applications such as food, nutraceutical, and adhesives, among others. Bananas provide a viable alternative to conventional sources of starch like corn due to their greater content of starch and resistant starch in addition to increased productivity and biomass [2]. Despite the high starch content in banana, it has not been adopted as a commercial starch; thus, it remains relatively low in commercial value. This could be attributed to the high variation in starch content across varieties and the high costs of production due to additional processing steps such as the prevention of browning. This generally limits the cost effectiveness of processing banana starch and commercial viability compared to traditional starches. The value of banana starch can be improved further by utilizing its functional and unique nutritional characteristics in the form of specialty secondary products such as resistant starch [5].

Nutritionally, starch is divided into three fractions based on degree of digestibility: readily digestible (RDS), slowly digestible (SDS), and resistant starch (RS) [6]. The ability of starch to resist digestion is largely influenced by the nature of the association between starch polymers, with higher amylose levels in the starch being associated with slower digestibility rates [6]. In raw foods such as banana and potato, starch is present as crystalline granules with two main forms, A and B, as classified using X-ray diffraction and Fourier transform infrared (FTIR) spectroscopy [6]. A-form starches have chain lengths of 23–29 glucose units and are usually found in cereals, while amylose-rich starches are B form with 30–44 glucose units [7]. Both forms have similar molecular arrangements with left-hand parallel double helices, but the B form has more associated water. A third form (C) is commonly found in legumes and is a mixture of A and B forms that resists digestion, as do B-form starches [7].

RS is also grouped into five types depending on the source and chemical modifications. Resistant starch Type 1 (RS1) and Type 2 (RS2) occur naturally in the grains and granules of bananas and potatoes, respectively. Type 3 (RS3) is retrograded amylose, while Type 4 (RS4) is chemically modified and Type 5 (RS5) arises due to the formation of amylose-lipid complexes. A substantial percentage of banana starch is RS, which is not digested in the small intestine but passes to the colon where it is fermented, releasing short-chain fatty acids (SCFAs), hydrogen, carbon dioxide, and methane [8], by the microflora. Resistant starch can, therefore, be defined as the sum of starch and products of starch degradation not absorbed in the small intestine of healthy individuals [9]. Butyrate, which is predominantly produced from RS2 fermentation, is the metabolic fuel of choice for colonocytes [10], while RS3 mainly gives acetate [11]. In addition to being a metabolic fuel for colonocytes, SCFAs mediate the beneficial effects of RS. These include improving blood flow in the colon, lowering luminal pH (since they are the principal anions), and preventing the development of abnormal colonic cell populations [7]. Overall, SCFAs have the potential to protect against colonic cancer [12,13,14]. RS benefits lipid and glucose metabolism, making it valuable for managing non-communicable diseases (NCDs) such as heart disease, diabetes, and obesity. RS significantly lowers plasma cholesterol, triglyceride, and lipoprotein levels [15].

Additionally, RS slows glucose release, reduces insulin response, increases fat utilization, and enhances satiety, aiding in diabetes and weight management [15]. Moreover, it also increases intestinal viscosity, impeding glucose absorption, and, thus, has a low glycemic index for extended glycemic control [16,17]. Combined with glycemic control, butyrate, the major SCFA from RS2 fermentation, elicits health benefits, including lowering metabolic markers for cancer, cardiovascular disease, and Type 2 diabetes and promoting a healthy gut, thereby preventing the advance of corresponding NCDs [18,19]. Resistant starch is also a prebiotic; therefore, it plays a major role in the nutraceutical industry, conferring health benefits upon consumers [20].

Natural variations in resistant starch content occur due to factors such as environmental conditions and mutations. Conditions such as temperature alter the activity of starch biosynthetic enzymes [21], impacting the thermal and digestibility properties of starches [22,23] and giving rise to variation in RS content. Additionally, starch structure varies in plants where genetic variation exists within botanical sources due to allelic variation in starch biosynthesis genes. In *Zea mays*, mutations at the amylose extender (ae) locus, which encodes the starch branching enzyme 2b, results in a highly resistant starch with higher apparent amylose content [24] than the normal ae germplasm. Studies by Vatanasuchart et al. [25] showed variation in resistant starch from Kulai cultivars from Thailand, signaling even greater variation within the vast biodiversity of bananas grown in the East African region.

The East African community (EAC) grows a variety of bananas with the potential to be utilized beyond just food in industrial products. However, a significant proportion of all bananas harvested are wasted due to perishability and low value-to-bulk ratio, causing high post-harvest losses. Value addition through industrial processing could provide a means to minimize losses and increase the market share of the crop. One such application would be as resistant starch, which would be a more economically viable alternative to native banana starch that cannot compete with cheaper traditional starches such as corn and cassava. Despite the health benefits and advantages of RS, there is limited information on the content, structure, and properties of RS in the commonly grown and consumed banana varieties in the EAC’s vast banana biodiversity. To aid the commercialization of RS, this study profiled the RS2 from the economically important banana varieties for the selection of the ultimate raw materials. 

## 2. Materials and Methods

### 2.1. Materials

Economically important banana cultivars native to Uganda were purposively selected, as listed in Table 1.

Green pre-climacteric bananas from each banana utilization grouping, with representatives from each of the five clone sets of EAHB described by Karamura et al. [26], were collected from the Banana Resource Centre, National Agricultural Research Laboratories (NARL)-Kawanda (1210 m, 0°24′ N and 32°31′ E) and analyzed at NARL-Kawanda. For the RS assays, positive and negative controls included RS from Megazyme and a native pure maize starch (Megazyme), respectively. The resistant starch, total starch, and amylose assay kits were purchased from Megazyme (Bray, Co., Wiklow, Ireland). The enzymes used in the experiments were purchased from Sigma-Aldrich, Co., (St. Louis, MO, USA). All chemicals were molecular and analytical grade purchased from Merck KGaA (Darmstadt, Germany), while the KBr used in FTIR was spectrophotometer grade, purchased from HiMedia Laboratories LLC (Kennett Square, PA, USA).

### 2.2. Isolation of Resistant Starch

Total starch was extracted from the green bananas using the alkaline method described by Evans [27]. Briefly, fruits were sliced, mixed in a 1% sodium bisulfite solution, and pulverized at low speed in a blender. The homogenate was consecutively sieved through stainless-steel mesh screens (Gilson, Lewis Center, OH, USA), the smallest being 180 µm, and progressively washed with distilled water and ethanol. The white-starch sediment was then dried in an oven (GENLAB, Cheshire, WA8 0SR. UK) at 40 °C for 48 h, and ground with an analytical mill (IKA, Janke & Kunkel-Str., Staufen, Germany) to a fine powder.

Resistant starch was isolated from the starch powder following the method of Wang et al. (2017) [28] with modifications. Briefly, extracted starch (1000 ± 5 mg) was weighed directly into each screw-cap culture tube. Pancreatic α-amylase (10 mg/mL), containing amyloglucosidase (AMG) (3 U/mL), was added to each tube and the tubes were incubated in a shaking water bath at 37 °C for 16 h. The reaction was terminated by adding 99% *v*/*v* ethanol. The tubes were centrifuged (MIKRO 220 R, Hettich AG. Bäch, Switzerland) at 1500× *g* for 10 min, followed by carefully decanting supernatants. The resultant pellets were dried to powder in an oven at 50 °C. The resultant powder (RS2) was stored for not longer than a month in sealed glass jars at room temperature and used for the proceeding experiments.

### 2.3. Total Starch, Polymer, and Resistant Starch Quantification

Total starch in the RS was determined using AOAC Method 996.11 (Megazyme). Results were expressed as percentage yield of the sample analyzed on a dry weight basis.

The RS content was determined by the K-RSTAR assay protocol (Megazyme). The absorbance of each solution was measured at 510 nm against a blank reagent using a spectrophotometer (Synergy2, BioTek^®^ Instruments, Winooski, VT, USA). The percentage resistant starch (dry weight basis) in the test samples was calculated from the equation
$$RS\left(\%\right)=\Delta E*\frac{F}{W}*90$$
where ΔE = absorbance (reaction) read against the reagent blank. F = conversion from absorbance to micrograms, W = the weight of the sample, and 90 = 162/180, which is the factor to convert from free D-glucose as determined to anhydro-D-glucose, as occurs in starch, expressed as a percentage.

Amylose and amylopectin content in the RS was determined using K-AMYL 06/18 protocol (Megazyme), and true amylose was expressed as a percentage of the dry weight. Using the difference from the total starch in the RS, the amylopectin fraction was determined.

### 2.4. Structural and Morphological Analysis

The samples of RS were suspended in 50% glycerol, and the morphology of granules was analyzed under an upright microscope (Zeiss Primostar3, Carl Zeiss AG, Oberkochen, Germany) at 400X magnification. The micro images captured with MotiCam2 were analyzed using ImageJ software (version 5.3.2e).

For SEM analysis, the RS samples were fixed on a double-sided adhesive carbon tape mounted on an aluminum stub. The samples were then coated with chromium using a Q150T Plus Turbomolecular pumped coater (Quorum Technologies. East Sussex, Laughton, UK) to a thickness of 1 nm and photographed using a Scanning Electron Microscope (VP Sigma 300 Zeiss, Carl Zeiss AG, Strasse 22, Oberkochen, Germany) in secondary electron mode at an accelerating potential of 10 kV. The images were further analyzed using ImageJ software (version 5.3.2e).

### 2.5. Analysis of Functional Properties

The swelling power of the RS was determined by the difference in weights of cooked starch and native starch according to the method of Singh et al. [29] with a few modifications. Briefly, 40 mL of a 1% starch suspension (*w*/*v*) was prepared in a 50 mL centrifuge tube. A magnetic stir bar was placed in the tube and the tube was placed in a water bath for 5 min at the constant temperature values of 50, 60, 70, and 80 °C, then kept at 90 °C for 30 min. The suspension was then centrifuged at 4000 rpm for 15 min, the supernatant was decanted, and the swollen granules were weighed. A 10 mL aliquot was taken from the supernatant, placed in a crucible, and dried in an oven at 120 °C for 4 h to constant weight. Percentage solubility (S) and Swelling Volume (SP) were calculated from the equations below:$$Solubility\left(\%\right)=\frac{weight\;of\;dried\;supernatant}{weight\;of\;sample(db)}\times 100\%$$
$$Swelling\;Volume\left(mL/g\right)=\frac{total\;volume-supernatant\;volume}{weight\;of\;sample(db)}$$

Water absorption capacity (WAC) was measured following the method described by Marta et al. [30]. Up to 10 mL of distilled water was added to 1 g of sample in a centrifuge tube and vigorously mixed using a vortex mixer. The sample was then conditioned at room temperature (25 ± 1 °C) for 1 h and then centrifuged (MIKRO 220 R, Hettich AG. Bäch, Switzerland) at 3500 rpm for 30 min. The volume of supernatant was measured and WAC was calculated using the equation below:$$WAC\left(\frac{g}{g}\right)=\frac{volume\;of\;water\;absorbed}{weight\;of\;sample\left(db\right)}$$

The pasting properties of the RS samples were analyzed using a Rapid Visco Analyzer (RVA 4500, PerkinElmer, Shelton, CT, USA). A 3.0 g sample was transferred onto the surface of 25 mL of distilled water in a test canister, after which the paddle was placed into the canister. Standard heating program 1 was used for the analysis, at a speed of 160 rpm for the blade rotation. The temperature profile was established at room temperature and during heating at 50 °C (0–1 min), heating from 50 to 95 °C (1–4.5 min), constant temperature at 95 °C (4.5–7.15 min), cooling from 95 to 50 °C (7.15–11 min), and constant temperature at 50 °C (11–13 min). The viscosity profile and parameters were calculated using the OriginPro software for Windows version 2021b (OriginLab Corporation). The viscosity was expressed as centipoise (cP). The analyzed parameters were pasting temperature (PT), peak viscosity (PV), holding viscosity (HV), breakdown (BD), final viscosity (FV), and setback (SB).

The clarity of a 1% solution was determined following the methodology by Craig et al. [31]. Briefly, a 1% aqueous starch solution was prepared by mixing 1.0 g of RS with 99.0 g of water. The mixture was then heated in a boiling water bath for 15 min under continuous stirring to obtain homogeneity. The solution was cooled to room temperature and absorbance was read at 620 nm using a spectrophotometer (Synergy2, BioTek^®^ Instruments, Winooski, VT, USA) over a 5-day period. Double-distilled water, having a transmittance (%T) of 100%, was used as a blank. Percentage transmittance, which is a measure of clarity, was calculated from the formula
Absorbance=2−log⁡(%T)

### 2.6. Fourier Transform Infrared Spectrometry

The degree of molecular organization was determined using FTIR. Infrared spectra were recorded using a Shimadzu IRAffinity-1S spectrometer (Shimadzu, Kyoto, Kyoto, Japan), following the methodology by Pozo et al. (2018) [32]. Briefly, 4 mg of the sample was mixed with 400 mg of spectrophotometer grade KBr and pressed into a pellet under 80 kN pressure using the hydraulic press. The pellet was held under pressure for 10 min for conditioning and then measured, and the FTIR spectra were analyzed in the range of 4500–500 cm^−1^ at a resolution of 2 cm^−1^ using the Happ–Genzel apodization function recorded from 50 scans. The raw spectral data were exported into OriginPro (2021b) for processing and curve fitting. Mathematical treatments were used to process the spectra: the Savitzky–Golay method at 30 points of window and a polynomial order of 2 was used to remove noise and irregularities. The peaks were then selected using the Gauss function and Levenberg Marquardt iteration algorithm to fit until convergence.

The ratio of intensity of the absorbance bands 1045/1022 cm^−1^ and 995/1022 cm^−1^ were used to quantify the differences in the degree of double helix and the degree of the molecular order of the RS polymers.

### 2.7. Statistical Analysis

The experiments were carried out in triplicate and are expressed as mean ± standard deviation. Data were assessed by one-way analysis of variance (ANOVA), and significant differences were identified using Tukey’s test. Inter-relationships between RS components and characteristics were analyzed by means of the Pearson correlation coefficient. All statistical analyses were performed using GraphPad Prism 8.0.2, while multivariate analysis was performed using OriginPro (2021b).

## 3. Results and Discussion

### 3.1. Resistant Starch Content and Polymer Composition

The content of RS observed in the selected banana cultivars ranged between 30.23 ± 3% and 79.27 ± 3.2% on a dry matter basis, as presented in Table 1. This result was consistent with previous studies that recorded values from 50% in Kluai cultivars [25] to 80% in the ABB genotype [28]. The ABB genome represented by the plantain genotypes cv Kivuvu and cv Kayinja had the highest content of RS, while cv Kibuzi from AAA-EA cooking/beer banana group had the lowest content. The percentage polymer composition of the RS (Table 2) shows amylose content ranging from 9.23 ± 0.4% to 22.39 ± 0.9%. Further, the linear relationship between the total amylose and the RS contents in the samples was not significant (r = −0.1062).

The amylose contents of the RS in the different banana cultivars were similar to those reported by Wang and Chen [28]. Collectively, the amylose content of the AAA genotype (11.16 ± 0.3%) was lower than that in AAA-EA (16.85 ± 0.01%), AAB (17.75 ± 0%), and ABB (22.39 ± 0.9). These results followed the same trend but were lower than those reported by Utrilla-Coello et al. [33], who found amylose content for *Musa* AAA (19.32–21.99%), *Musa* AAB (26.35%), and *Musa* ABB (25.38%). The amylose concentration of banana starches varies across studies: 25.8–35.5% in the Kluai cultivars [34], 10–11% for plantains [35], and 40.7% for cv Valery [36]. Correlation analysis showed no significant linear relationship between the total amylose and the RS contents in the banana samples (r = −0.1062). The results compare well with findings by Vatanasuchart et al. [34], whereas an earlier study by Eerlingen and Delcour [37] indicated a positive correlation between amylose content and RS where they reported high-resistant starch involving amylose crystallization, particularly in rice.

### 3.2. Structural Analysis

#### 3.2.1. Granule Size and Distribution

The analysis of granule size and distribution was performed for RS from all the banana varieties/genotypes tested. The frequency distributions of the granule diameters for each variety are represented by smooth curves in Figure 1. The mean diameters of the granules ranged from 17.59 ± 7.0 µm in cv Kibuzi to 30.04 ± 6.5 µm in cv Kivuvu, as given in Table 2. The granule size of banana RS across all the cultivars was significantly different (*p* < 0.0001). RS from cv Kibuzi was populated with small round granules, which are easily hydrolyzed by enzymes, as reported by Soares et al. [38], explaining its low RS content. The reverse is true for cv Kivuvu, with comparatively larger granules > 30 µm, which provide a smaller surface area-to-volume ratio for enzyme activity, higher resistance to enzymatic hydrolysis, and, therefore, higher RS content.

#### 3.2.2. Granule Characteristics

Light microscopic imaging showed that the granules of banana resistant starch for all varieties had varying shapes and sizes, as shown in Figure 1. Majority of the RS granules from AAA and ABB genotypes were big, i.e., greater than 35 µm, interspersed with a few small ones with sizes detailed in Section 3.2.1. Such large granules were predominantly elliptic, but a small number were irregular in shape, as evidenced in the NARITA21 and KABANA6H micrographs. The RS from AAA-EA exhibited a mixture of irregularly shaped elongated, spheroid, and oval-shaped granules with generally smooth surfaces. High-magnification SEM micrographs revealed that RS granules from EAHB varieties like cv Nfuuka and cv Mpologoma particularly had smooth surfaces with no signs of degradation (Figure 2). The high-resolution SEM micrographs revealed folds, pits, and troughs in the micro-architecture of some of the RS granular surfaces. 

In addition to the size of the granules, the surface characteristics of the granules could also explain high RS content in cultivars like cv Nfuuka, which have relatively small granule sizes, as shown in Figure 2(A1). The high-magnification (SEM) images of the granule surfaces of hybrid cultivars showed the presence of pits and troughs that were approximately 100 nm–200 nm. This surface microarchitecture presents increased surface area-to-volume ratio for enzymatic degradation on the granules that have them, even though the mean diameter of the granule is actually large, for example, in dessert bananas varieties like cv Bogoya and cv Sukali Ndiizi and hybrids like NARITA21 and KABANA6H. This also indicates that ripening starts earliest in these cultivars [38]. The vestiges of starch breakdown by amylases both during the RS isolation and the banana plant intrinsic mechanism could be seen in the layer expositions at the apexes of some RS granules like Gonja (Figure 2(A2,B2)).

### 3.3. Functional Properties

#### 3.3.1. Clarity (Transmittance)

The percentage transmittance, which is a measure of clarity for the RS gels, is shown in Appendix A in the Appendix A. The percentage transmittance was higher at hour zero, but reduced as the amylose leached into solution over the span of the experimental time, thus lowering the transmittance. Among the samples analyzed, RS from Sukali Ndiizi, Kibuzi, and FHIA17 differed from this trend by showing increased transmittance with time. The gels from Sukali Ndiizi had a significantly higher clarity than others throughout the experimental period. Sukali Ndiizi and Kibuzi recorded the highest final transmittance at 93%, and this could be attributed to the loosely packed granule structure and small granule size, respectively; thus, the amylose leached easily.

In addition to being a test for starch purity, paste clarity is also one of the most essential functional characteristics of starches in food products [39]. Since thickening agents in juices, soups, and pie fillings should be made from transparent starches [40], RS from cv S. Ndiizi and Kibuzi would be excellent for this alternative application.

#### 3.3.2. Pasting Properties of Banana RS

When heated in excess water, the granules underwent molecular changes via a series of stages that are illustrated by their pasting properties (Figure 3). Such pasting profiles estimate starch–water binding capacity and the strength of bonds in the starch granule; therefore, they can be used to predict starch processing qualities. All the banana RS, except that from FHIA17 and NAROBAN5, exhibited a hybrid of A and B viscosity curves, characterized by a high peak viscosity, followed by a major breakdown and, finally, a high final viscosity (Figure 3). The profiles obtained were comparable to those reported in native banana starches reported by [39,41,42] since no modification was made during processing. The variation in pasting temperature (PT) among the samples was not significant while peak viscosity (PV) varied significantly among the varieties (Appendix A of the Appendix A). The observed variation in peak viscosity could be due to different characteristics such as granule shape, size, and amylose content [43]. The PT and PV values were associated with the ability of starch granules to bind water at the rate at which the granules disintegrated. Since the swelling ability of the granules was similar between 70 and 80 °C, which is the pasting temperature, there was no significant difference between the cultivars in this property. Hybrid KABANA6H, Figure 3C, had the lowest breakdown at 901.5 ± 14.9 cP but a relatively high setback viscosity at 1240 ± 169 cP. Interestingly, the peak did not form fully throughout the duration of the experiment, with peak viscosity being recorded at only 4.1 min. The KABANA6H and NAROBA5 hybrids, Figure 3C, were characterized by significantly lower breakdowns at 901.5 ± 14.9 and 984 ± 0 cP, respectively. The increase in the setback viscosity of the RS was correlated to the increase in solubility (r = 0.5508, *p* < 0.0219), which indicated an increase in the leaching of amylose. The pasting results suggest that banana RS would have high paste stability in mechanical and hydrothermal processes such as cooking. While the pasting properties indicate that the RS remains stable in cooking processes, its viscosity increased when cooled, indicating instability during cooling processes.

#### 3.3.3. Swelling Power and Solubility

The swelling stage of starch granules is the initial step of all other paste characteristics. The swelling capacity of starch granules and their corresponding solubility (S) as they pass through different stages from water absorption to granule disintegration is a pre-eminent structural characteristic. Given the very similar swelling profiles, an equally similar pattern was found for the solubility of the RS across samples. Figure 4B shows the changes in the swelling power (SP) of granules as a function of temperature for each variety. The onset of granular swelling for most varieties was at 60 °C and showed significant SP at 70 °C. An exponential increase in swelling was observed between 70 and 80 °C, which is associated with the peak pasting temperature. At 90 °C, RS from cv Kivuvu presented the highest swelling value, but no remarkable differences were observed between the different varieties at this temperature. As expected, swelling power increased with increments in the pasting temperature due to amylose solubilization during starch gelatinization at high temperature and, therefore, contributed to the swelling of the granules. In their study, Bello-Pérez et al. [44] reported similar behavior and values of swelling for the banana starches.

Amylose was of great importance in the initial resistance of granules to swelling and solubility, as swelling occurs quickly after amylose molecules are leached [45]. Additionally, the increase in setback viscosity of the RS was correlated to the increase of solubility (r = 0.5508, *p* < 0.0219), which indicated an increase in the leaching of amylose, as elaborated in Section 3.3.2.

The solubility of the banana RS increased with increase in temperature (Figure 4A). At 90 °C, only NARITA21 had a significantly higher solubility compared to the other cultivars. Regardless of solubility, an increase in the swelling of the granules was observed. However, after 70 °C, there was a reduction in solubility, probably due to the disruption of the granule structure and the leaching of compounds in the granules, according to the Hoover study [46]. For all the banana RS samples, the water absorption capacity, swelling power, and solubility were directly correlated to the increase in temperature. The swelling patterns revealed that, at temperatures below 70 °C, the granules did not significantly swell, which suggested that strong forces acted between the molecules, which maintained the structural integrity of the granules. Studies by Bello-Pérez et al. [44] demonstrate how the granules inflated rapidly at temperatures above 70 °C due to the breakage of the intermolecular hydrogen bonds in amorphous portions, allowing irreversible and progressive water absorption. A similar swelling profile was observed for all samples of RS from the different banana cultivars studied. The higher SP of RS from cv Kivuvu was directly related to the amylose–amylopectin bonding level as well as its molecular association with intramolecular water, evidenced by the FTIR analysis. 

### 3.4. Molecular Organization of the Resistant Starch 

From the FTIR analysis, it was evident that all the samples of RS from banana had a similar chemical structure. The exact positions of the peaks used for band assignment are given in Appendix A in the Appendix A. Reports by Abdullah et al. [47] and Paramasivam et al. [2]. were the basis for band assignment. The strong peak at 3000–3600 cm^−1^ was due to the stretching vibration of the OH groups in the starch. The peak at 2931 cm^−1^ was due to the stretching vibration of the saturation of C-H. The absorption peak between 1637 cm^−1^ and 1648 cm^−1^ was due to the C=O bending associated with the OH group. The absorption peaks arising from the asymmetric stretching of the C-O-C and C-O bonds were at 1149 cm^−1^. The band between 1200–800 cm^−1^ was a result of C-O stretching vibration in the pyranose, while the 920 cm^−1^ to 711 cm^−1^ was due to C-O-C ring vibration in the D-glucopyranose.

The varieties studied showed the typical FTIR spectra of A, B, and C polymorphs of starch described by [32], as shown in Figure 5. The basis of the differentiations into A, B and C types was the intensity of the OH band, which was higher in the A-type than B- and C-types, due to the higher density of strong hydrogen bonding interactions in A-types [48]. The other differentiating characteristic was the width at half the maximum height of the OH bands, with the width of the B-type being higher than A-type and C-type. The FTIR spectra of starch are susceptible to structural alterations at the short-range molecule level often depicted as double helices, with the absorbance bands at 1022 and 1045 cm^−1^ being indicative of both the ordered and amorphous structures that make up the starch’s crystalline region [49]. Although Liu et al. [50] reported that the crystal morphology is the same in the same crop, the banana varieties studied exhibited different polymorphs. These differences in crystalline structure are attributed to variation in the packing density of double helices and the structural water content. The B-type polymorph holds much more structural water than the A-type [48]; hence, the double helix strands that form crystallites associate more strongly with the extra hydrogen bonds in the water molecules. This is evidenced by the broader peaks in the 800–1300 cm^−1^ and the OH region. The quantification of the peak ratios 1045/1022 cm^−1^ and 995/1022 cm^−1^, as per Zhang et al. [51], yielded differences in the degree of double helix and the degree of the molecular order of the RS polymers, respectively (Appendix Ain the Appendix A). These incongruent values are explained by the contribution of the structural water being higher in the B-type polymorph than that in A-type [32].

The results of PCA for the FTIR spectra (Figure 6) derived from the loading plot (Appendix A of the Appendix A) indicated that 89.6% of the variation can be explained by PC1 and 5.9% by PC2. Two distinct groups could be identified based on the variation in these two areas of the FTIR spectrum. The varieties with B-type RS grouped in the upper right quadrant, while the A-type was in the left quadrants, with C-type dispersed between the two. The grouping could be attributed to the different distribution and composition of A and B polymorphisms in the C-type starches studied. In related studies, a similar distribution in grouping was observed in starches studied by Pozo et al. [32]. According to their characteristic FTIR spectral signatures, an unambiguous classification of the varieties was possible, enabled by their distinctive OH and crystalline regions. This underlines that FTIR spectroscopy in the fingerprint region combined with chemometric data analysis presents a powerful tool for the reliable discrimination between RS from different banana varieties.

### 3.5. Relationships between the Characteristics of Resistant Starch

Correlation analyses were performed to explore the underlying relationships between compositional characteristics and physicochemical properties of resistant starch (Figure 7). RS was positively correlated (r = 0.58, *p* < 0.0149) with granule size (GS) and gelatinization temperature (Tg) (r = 0.65, *p* = 0.0068), but not with amylose content (AM) (r = −0.11, *p* = 0.68499). The granule size is key in resisting enzymatic hydrolysis; equally so is the microarchitecture. The higher surface area to volume ratio presented by the small granules such as those from cv Kibuzi (<18 µm) makes them easier to hydrolyze; hence, a lower RS content is assayed. However, the bigger granules such as those from cv Kivuvu present a lower surface area to volume ratio for enzyme action. In addition to the size, the surface microarchitecture could influence the RS content assayed, as explained in Section 3.2.2. As shown by the pasting temperature (PT), the bigger granules also took longer to absorb water at even higher temperatures in order to reach their maximum gelatinization, as opposed to the smaller granules. These PT and PV values were inversely associated with the ability of granules to bind water and with the rate at which the granules disintegrated. The strong positive association observed between GS and clarity (r = 0.71, *p* < 0.01493) is attributed to the bigger granules leaching low levels of amylose much more slowly throughout the length of the experiments. 

In contrast to GS, amylose displayed remarkably inverse correlation patterns with other variables, exhibiting a strong negative association to GS, swelling power (SP), and peak viscosity (PV) (r = −0.68, −0.89 and −0.67, *p* < 0.00247, *p* < 1.24948 × 10^−6^, *p* < 0.00301), respectively, but had a positive correlation to pasting time (r = 0.63, *p* < 0.00697). Strong negative correlation (r = −0.82, *p* < 5.1852 × 10^−5^) was also detected between AM and clarity. Amylose is of great importance in the initial resistance of granules to swelling and solubility, as swelling occurs quickly after amylose molecules are leached [45]. The leached amylose absorbs most of the light passing through the starch solution during spectrophotometry; thus, low transmittance/clarity of high-amylose cultivars is observed. 

Agglomerative hierarchical clustering (AHC) based on correlation between parameters (Appendix A in the Appendix A) showed that the groupings generated from the inter-relationships between the biochemical, structural, and functional characteristics of the RS across cultivars can be the basis for the selection of specialized RS for industrial applications. For instance, group 1 (G1), populated by varieties with high RS content, would be suitable for nutraceuticals such as a high dietary fiber supplements. In addition, the RS from these varieties would also be appropriate for the development of extruded foods with low and moderate moisture such as ready-to-eat cereals. It would not only provide a good amount of dietary fiber in the product, but would also provide excellent texture without compromising quality.

## 4. Conclusions

The banana varieties studied vary in RS content, relative crystallinity, granule size distribution, and microstructure. As signified by the correlation results, granule size alone is not responsible for the variation in starch digestion and the resulting RS content. The granule distribution within the RS and the microstructure equally influence the overall digestibility of the granules and thus the RS content of a particular variety. The results demonstrated that RS2 from bananas varies in structure and functionality; therefore, the utilization and processing effects of RS from each variety need to be considered individually. The high-resistant starch varieties such as the G1 cultivars could be utilized for the development of nutritional supplements or extruded functional foods. The RS from S. Ndiizi and Kibuzi would be excellent for alternative applications requiring transparent starches such as juice thickening agents. This study, therefore, provides fundamental information on the suitable banana varieties to use for the production of RS-based commercial products.

## Figures and Tables

**Figure 1 foods-13-02998-f001:**
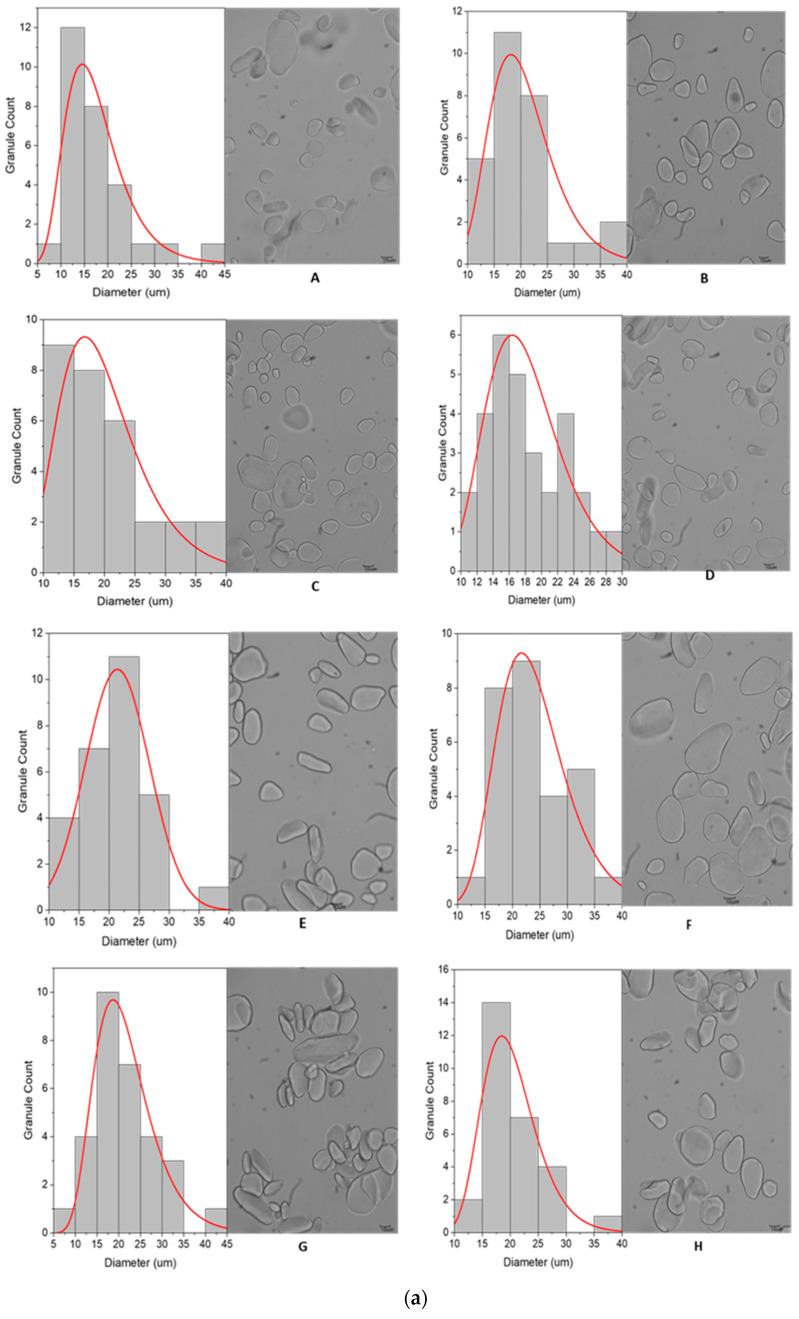
(**a**): Micrograph and granule distribution of granules in the analyszd banana varieties. A—cv Kibuzi, B—cv Mpologoma, C—cv Nakitembe, D—cv Nfuuka, E—cv Mbidde, F—FHIA17, G—KABANA6H, H—NAROBAN5. (**b**): Micrograph and granule distribution of granules in the analyzed banana varieties. I—NARITA21, J—NARITA23, K—cv Bogoya, L—cv Ndiizi, M—cv Gonja, N—cv Kivuvu, O—cv Kayinja.

**Figure 2 foods-13-02998-f002:**
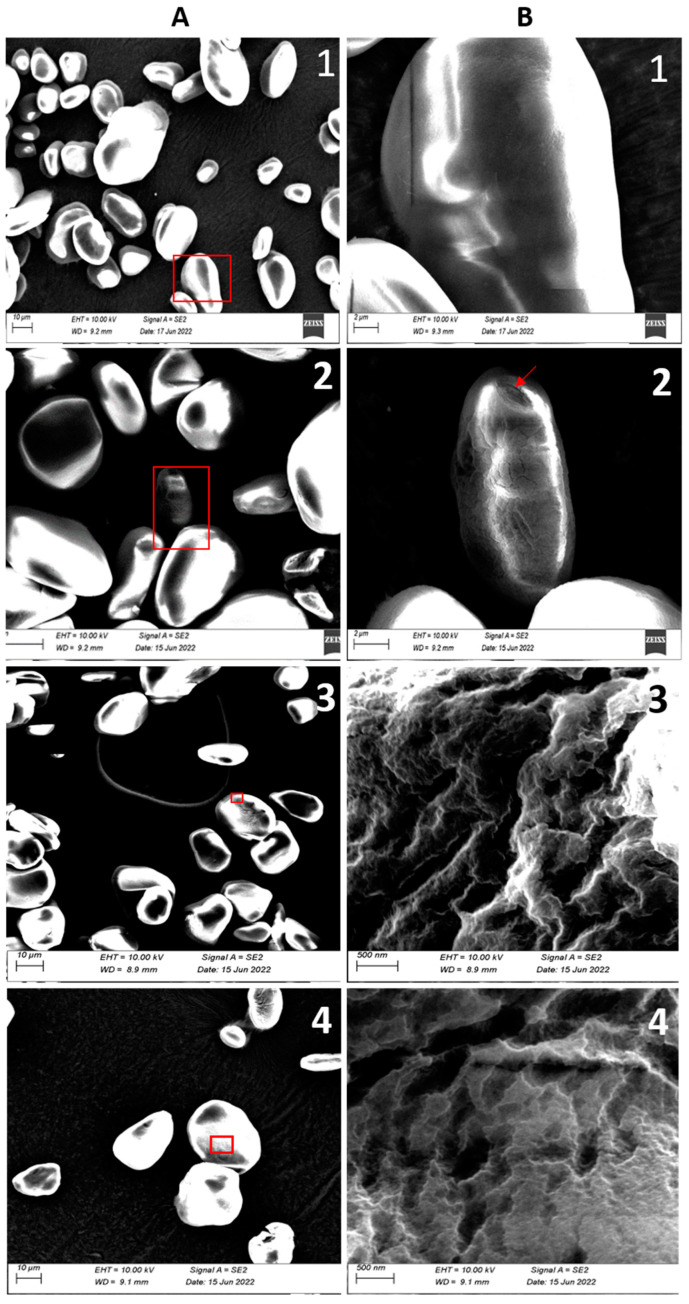
SEM imaging showing granule shape and size at EHT = 10 kv and WD = 9.2 mm, Scale bar: 10 µm (**A**). The high-magnification imaging from the red insets shows the micro-architecture of the granules (**B**). 1 depicts smooth granules of cv Nfuuka; 2 shows the layer expositions at the apex of cv Gonja, scale bar: 2 µm; 3 shows the folds of peaks and troughs in NARITA21; and 4 shows the grooves and pits in KABANA6H, scale bar: 500 nm.

**Figure 3 foods-13-02998-f003:**
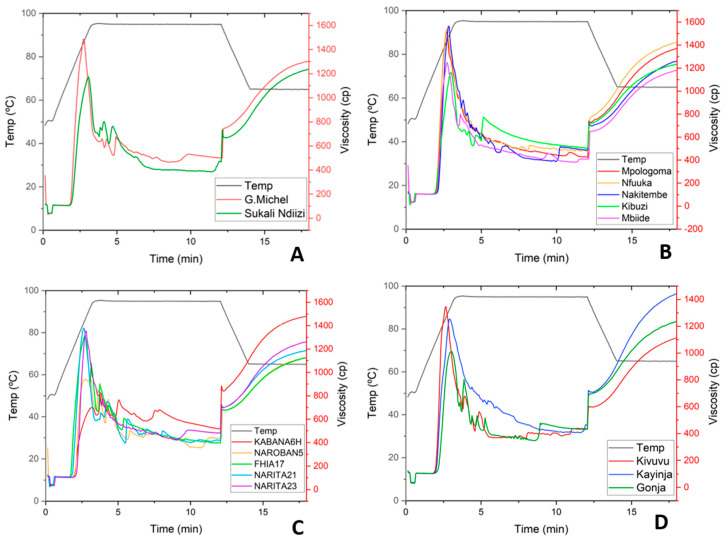
Viscosity profiles for resistant starch from banana varieties: (**A**) Dessert, (**B**) Cooking-EAHB, (**C**) Hybrids, and (**D**) Plantain.

**Figure 4 foods-13-02998-f004:**
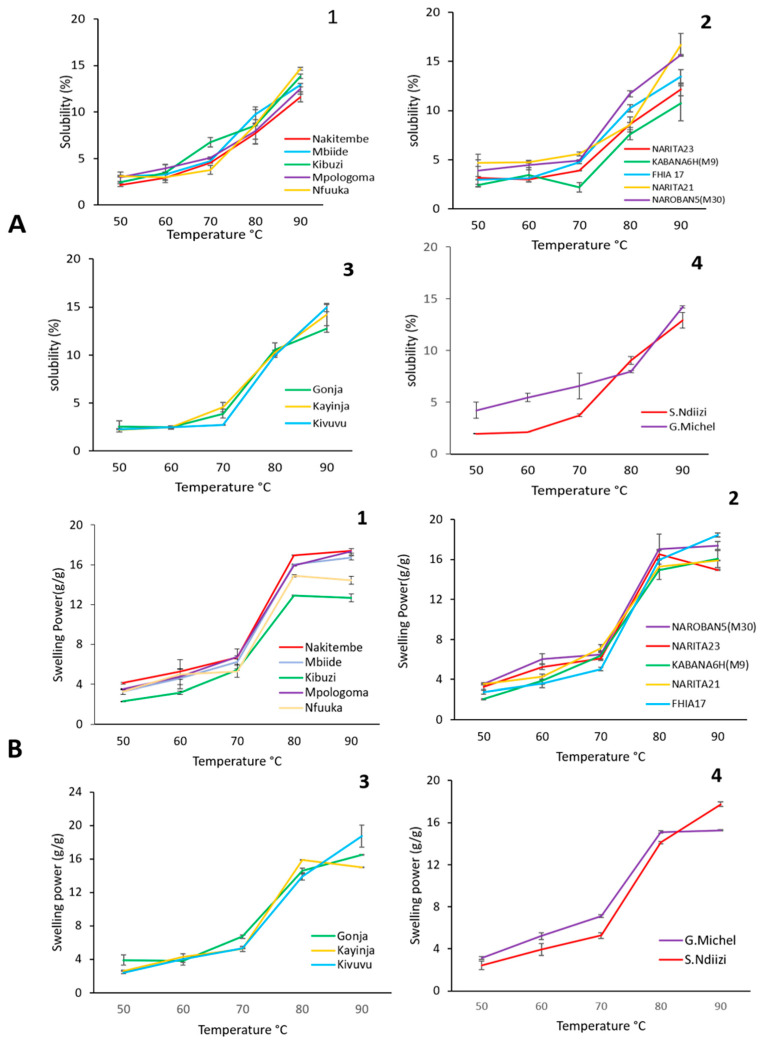
Swelling profiles (**A**) and solubility profiles (**B**) of 1% (*w*/*v*) starch suspension at 50–90 °C, showing 1—Cooking-EAHB, 2—Hybrids, 3—Plantain, 4—Dessert banana varieties.

**Figure 5 foods-13-02998-f005:**
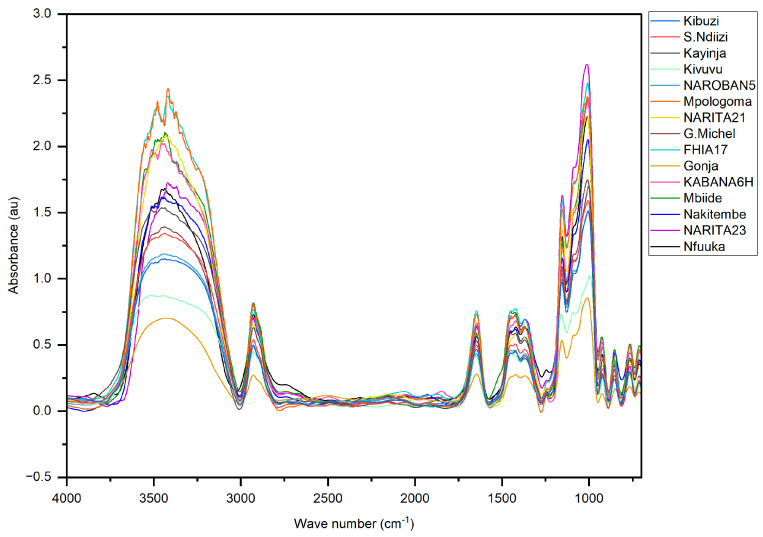
FTIR spectra of the RS samples from banana analyzed in the range of 4500–500 cm^−1^. Inset: absorption peaks used to determine the degree of crystallinity.

**Figure 6 foods-13-02998-f006:**
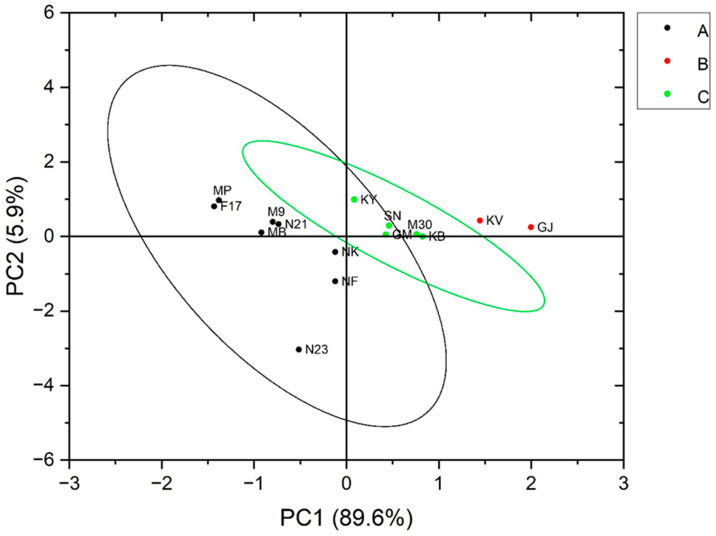
Biplot of the principal component analysis PC1 vs. PC2, describing the scores and variation in the crystallinity of the samples analyzed, showing scores represented by 95% confidence ellipses. MP—Mpologoma, MB—Mbiide, NK—Nakitembe, NF—Nfuuka, KB—Kibuzi, KY—Kayinja, KV—Kivuvu, GJ—Gonja, SN—Sukali Ndiizi, GM—Gros Michel, M30—NAROBAN5, M9—KABANA6H, N21—NARITA21, N23—NARITA23, F17—FHIA17. A—Samples display the A polymorph, B—samples display the B polymorph, and C—samples display a mixture of A and B, as described by the Pozo study [32].

**Figure 7 foods-13-02998-f007:**
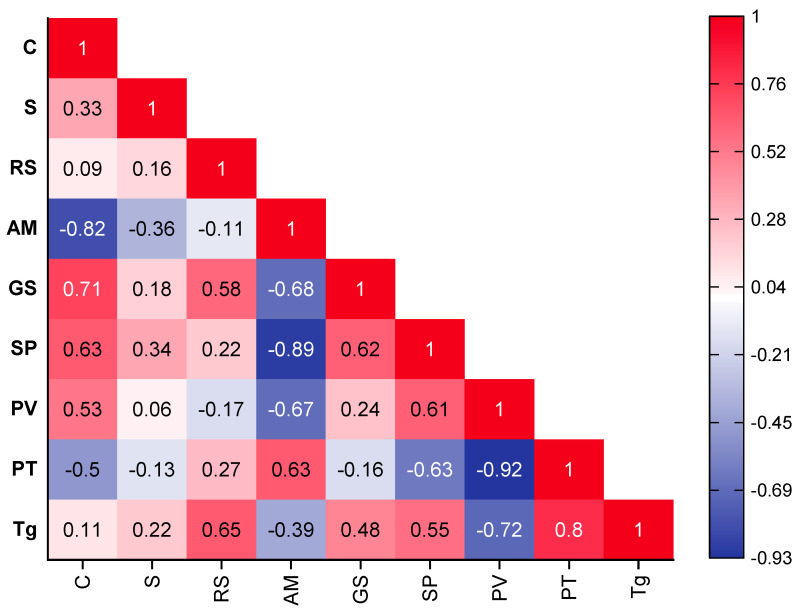
Heat-map displaying the extent and direction of correlations (r) between RS compositional characteristics and physiochemical properties. Correlations were statistically significant at r ≥ 0.49 and r ≤ −0.49. C—clarity/%transmittance, RS—resistant starch content (g/100 g), AM—Amylose content (%), S—solubility at gelatinization (%), Tg—Gelatinization temperature (°C), GS—median granule size (µm), SP—swelling power (%), PV—peak viscosity (cP), PT—pasting time (min).

**Table 1 foods-13-02998-t001:** Sampled banana cultivar groupings and genotypes.

	Cultivar/Variety	Genotype	Class (Utilisation)
1	Mpologoma	AAA-EA	Cooking
2	Nfuuka	AAA-EA	Cooking
3	Kibuzi	AAA-EA	Cooking
4	Nakitembe	AAA-EA	Cooking
5	Mbidde	AAA-EA	Brewing
6	Kayinja	ABB	Brewing
7	Sukali Ndiizi	AAB	Dessert
8	Gros Michel (Bogoya)	AAA	Dessert
9	Gonja	AAB	Plantain
10	Kivuvu	ABB	Plantain
11	NARITA21	AAA	Hybrids
12	NARITA23	AAA	Hybrids
13	KABANA 6H (M9)	AAA	Hybrids
14	NAROBAN5 (M30)	AAA	Hybrids
15	FHIA17	AAAA	Hybrids

**Table 2 foods-13-02998-t002:** Resistant starch content, polymer of composition, and granule size of the cultivars.

Sample	Genotype	RS Content (g/100 g) *	Amylose (%) †	Granule Size (µm)
Kayinja	ABB	79.27 ± 3.2 ^a^	22.4 ± 0.9 ^bc^	23.80 ± 6.3
Kivuvu	ABB	69.93 ± 2.8 ^ab^	13.34 ± 0.8 ^defg^	30.04 ± 6.5
Gonja	AAB	33.34 ± 2.9 ^gh^	17.7 ± 0.1 ^bcd^	18.93 ± 6.0
NARITA23	AAA	35.77 ± 0.3 ^fgh^	15.2 ± 0.7 ^defg^	20.54 ± 5.9
NARITA21	AAA	35.945 ± 0.4 ^fgh^	14.5 ± 1.1 ^defg^	21.12 ± 5.3
KABANA6H	AAA	44.33 ± 2.9 ^defg^	12.9 ± 0.1 ^defg^	21.63 ± 7.0
NAROBAN5	AAA	55.295 ± 1.1 ^cd^	9.59 ± 2.9 ^g^	20.22 ± 5.1
FHIA17	AAAA	45.48 ± 1.2 ^def^	9.23 ± 0.4 ^g^	24.09 ± 6.4
Mbiide	AAA-EA	65.91 ± 2.2 ^bc^	14.3 ± 2.9 ^defg^	21.39 ± 5.3
Nfuuka	AAA-EA	53.97 ± 3.3 ^d^	14.0 ± 0.5 ^defg^	18.18 ± 4.9
Nakitembe	AAA-EA	51.785 ± 0.9 ^d^	13.98 ± 2.1 ^defg^	20.13 ± 7.5
Mpologoma	AAA-EA	50.75 ± 0.5 ^de^	16.3 ± 0.3 ^cde^	20.69 ± 6.5
Kibuzi	AAA-EA	30.23 ± 3.0 ^h^	10.03 ± 0.2 ^fg^	17.59 ± 7.0
Sukali Ndiizi	AAB	29.3 ± 7.9 ^h^	12.9 ± 1.3 ^defg^	25.71 ± 5.2
Gros Michel	AAA	49.315 ± 0.1 ^de^	11.2 ± 0.1 ^efg^	18.45 ± 8.2

Values are presented as mean values ± standard deviation. Values that do not share the same letter in the same column differ significantly according to Tukey’s test (*p* < 0.05). * 100 g of native starch. † Amylose as a percentage of the total starch in the RS.

## Data Availability

The original contributions presented in the study are included in the article/Appendix A, further inquiries can be directed to the corresponding author.

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
