# Peer review of "Variation and Abundance of Resistant Starch in Selected Banana Cultivars in Uganda"

_foods, 2024, doi:10.3390/foods13182998_

Round 1
Reviewer 1 Report
Comments and Suggestions for Authors
In the present study, resistant starch was extracted from 15 different banana varieties, and compared their physiochemical and structural characteristics. Although authors tested a large number of samples, the experimental design was incomplete. The current data could not provide enough information for analyzing the structure of RS. Additionally, the introduction should be rewritten.
1. Authors stated “Despite the high starch content in banana, it has not been adopted as a commercial starch thus remains relatively low in commercial value” in Line 39-40, why??
2. Although authors provided a comprehensive introduction to resistant starch, it was mostly common knowledge. Additionally, authors did not address the necessity and significance of this research in the introduction.
3. In section 2.3, the equation was missing.
4. What is the type of extracted RS?
5. DSC and XRD measurements should be proceeded to analyze the structure of extracted starch.
6. In vitro digestion also should be proceeded.
Comments on the Quality of English LanguageOverall, the quality of English of this paper is good, but minor editing is still required.
Author Response
To the Editors,
Thank you for the opportunity to revise and resubmit our manuscript entitled, Variation and abundance of Resistant Starch in Selected Banana Cultivars in Uganda. We appreciated the helpful comments from the editors and peer reviewers. Below, please find our itemized responses to the comments to accompany the revised manuscript we are submitting.
Thank you for your further consideration of this paper for potential publication in MDPI-Foods.
Sincerely,
Ali Kajubi
Reviewer 1 Comments to Author: In the present study, resistant starch was extracted from 15 different banana varieties, and compared their physiochemical and structural characteristics. Although authors tested a large number of samples, the experimental design was incomplete. The current data could not provide enough information for analyzing the structure of RS. Additionally, the introduction should be rewritten.
Response: We believe we have provided sufficient data to answer the question of variability of resistant starch in bananas grown in Uganda. Even so, additional data was provided as supplementary material. The introduction has been rewritten highlighting the significance of the study and reducing what the reviewer referred to as “common knowledge”.
- Authors stated “Despite the high starch content in banana, it has not been adopted as a commercial starch thus remains relatively low in commercial value” in Line 39-40, why??
Response: Several challenges hinder their commercial viability as a starch source including Variability in Starch Content across varieties, more so when the bananas ripen thus presenting a short working time for high starch yield. Additionally, processing starch from bananas requires special conditions such as those preventing browning and yet such additional processing steps are not required to the traditional starch crops such as corn and cassava.
- Although authors provided a comprehensive introduction to resistant starch, it was mostly common knowledge. Additionally, authors did not address the necessity and significance of this research in the introduction.
Response: The introduction has been reworked highlighting the gaps and significance of the research
- In section 2.3, the equation was missing.
Response: The equation has been provided as , Line 155
- What is the type of extracted RS?
Response: The extracted RS is type 2 resistant starch as was stated in the writeup. Line 110 and 145
- DSC and XRD measurements should be proceeded to analyze the structure of extracted starch.
Response: We acknowledge that Xray diffraction is key in analysing the degree of crystallinity in starches through determining the long-range crystalline order in the starch related to the packing of the double helices. This is a limitation in the study, however, the study by (Pozo et al., 2018) highlighted an alternative methodology by comparing both FTIR and XRD. They concluded that although FTIR analysis did not provide enough information about the amount of large order starch structures (crystallinity). The FTIR technique was sufficient to differentiate starch polymorphs, a characteristic that affects the functional properties of the starch. It is against this background that we used the FTIR for deducing the starch polymorphs in the banana RS analysed and addressed the limitation. We also acknowledge that DSC is a very powerful tool in deducing gelatinisation and melting of the fat complexes and would indeed add value to our study. To gain insight into gelatinisation behaviour, we utilised functional properties involving hydrothermal treatments such as RVA to give insight into the behaviour of the granules in cooking systems.
- In vitro digestion also should be proceeded.
Response: The resistant starch studied is the resultant starch from digestion of native banana starch using α-amylase and amyloglucosidase. In vitro digestion would be relevant if we were studying native starches.

Reviewer 2 Report
Comments and Suggestions for Authors
The manuscript is well-written, scientifically robust, and utilizes advanced analytical tools. However, a limitation is the narrow scope of the research. This exploratory study investigates resistant starch (RS) in 15 banana varieties grown in Uganda, focusing solely on the cultivar as the variable. Other factors that may influence RS content in bananas, such as maturity stage, growing conditions, agricultural practices, and post-harvest handling, were not examined. As a result, the study does not provide comprehensive information that could be valuable for farmers and the food industry.
Introduction
Please provide a summary of existing research on resistant starch in bananas, highlighting gaps in the research, concerning the variation and abundance in Ugandan cultivars.
Please emphasize the potential applications of its findings in improving banana breeding programs, and food industry practices. Understanding the variation in resistant starch levels is crucial for health, agricultural, and industrial purposes.
Identify the factors influencing resistant starch levels and compare the resistant starch content with that of other regions or cultivars.
The equation on Line 158 is missing.
The authors must ensure true replication in this experiment by specifying the sampling method and detailing the data sources to determine their suitability for analysis using one-way ANOVA.
The authors should consider reporting the decimal points of the data in Table 1 and ensure consistency with the content in Lines 243-251.
In Lines 355-356, the authors mentioned that RVA profiles of RS and native starch were comparable without explanation; please provide a discussion on this point.
The correlation between granule shape, size, amylose content, and pasting properties should be discussed (Line 360).
The RVA parameters, such as pasting temperature, peak viscosity, setback, final viscosity, and breakdown, should be presented in a Table with statistical analysis.
Section 3.5 lacks discussion and should be elaborated upon.
Comments on the Quality of English LanguageThe manuscript is of good English quality and contains only a few errors.
Author Response
Introduction
Please provide a summary of existing research on resistant starch in bananas, highlighting gaps in the research, concerning the variation and abundance in Ugandan cultivars. Identify the factors influencing resistant starch levels and compare the resistant starch content with that of other regions or cultivars.
Thank you very much for pointing this out. The gaps addressed by the research have been pointed out such as the information gap on the properties of RS in the commonly grown Ugandan banana varieties to aid commercialisation.
Factors influencing variation in resistant starch such as natural variation due to environment and mutations have been discussed Line 89-99.
Please emphasize the potential applications of its findings in improving banana breeding programs, and food industry practices. Understanding the variation in resistant starch levels is crucial for health, agricultural, and industrial purposes.
Importance of utilisation of the resistant starch have further been emphasised in the introduction, Line 71-88 and in the conclusion Line 517-522.
The equation on Line 158 is missing.
Response: The equation has been provided as , Line 155
The authors must ensure true replication in this experiment by specifying the sampling method and detailing the data sources to determine their suitability for analysis using one-way ANOVA.
Experiments were carried out in triplicate, the relevant statistical analysis has been discussed in the exact section it was used, for example; ANOVA and a post hoc test carried out on all cultivars with each cultivar in 3 replicates of in table 2, Line 253.
The authors should consider reporting the decimal points of the data in Table 1 and ensure consistency with the content in Lines 243-251.
The decimal points have been standardised to 2 decimal places for consistence. Table 2. Line 253
In Lines 355-356, the authors mentioned that RVA profiles of RS and native starch were comparable without explanation; please provide a discussion on this point.
The correlation between granule shape, size, amylose content, and pasting properties should be discussed (Line 360).
The correlation results have been discussed and elaborated in section 3.5
The RVA parameters, such as pasting temperature, peak viscosity, setback, final viscosity, and breakdown, should be presented in a Table with statistical analysis.
This table has been provided as Table S3 in supplementary data
Section 3.5 lacks discussion and should be elaborated upon.
The correlation results in 3.5 have been elaborated upon citing reasons for the observed associations. Line 468-489

Reviewer 3 Report
Comments and Suggestions for Authors
See the attached manuscript for more detailed comments.
Abstract - I would suggest supplementing with the aim of this study.
2. Materials and Methods - I would recommend revising the layout of the manuscript (eg the layout of chapter and subsection headings).
2.1 Materials - I would recommend to supplement with the exact geographical coordinates of the banana growing regions. And in addition, this information is presented in the form of a table.
2.2 Isolation of Resistant starch - All equipment used must specify make, manufacturer, country. Check all equipment! Did the experiment take place at exactly 40°C? Maybe there was ± some degree? For example 40±1°C? Check all temperature! There are questions about the storage of this RS powder - in what package was it stored, where was it stored, under what conditions was it stored, how long was it stored?
2.3 Total starch, polymer and Resistant starch quantification - Where is the equation?
2.6 Fourier-Transform Infrared Spectrometry – Is this the correct notation "cm−1"?
2.7 Statistical analysis - How many repetitions were performed during the experiments? At what threshold were significant or non-significant differences found? "Principal component analysis" and “Cluster analysis” is also found in the manuscript, so this should be mentioned and explained in this section.
3. Results and Discussion –
3.2 Structural analysis
3.2.1 Granule size and distribution - The information I suggested should be given in the form of a table. It is very difficult to grasp now.
3.2.2 Granule characteristics - Where can I see "Figure 2-2."? Only "Figure 2 A 2" and "Figure 2 B 2" are found in this manuscript.
3.3 Functional properties
3.3.3 Swelling power and solubility - Maybe it's better to say "according to the Hoover study".
3.4 Molecular organization of the Resistant Starch - I would recommend adding a reference, as the authors in this study have hardly investigated and proven what type of links and between which elements are formed or broken. I would suggest adding an explanation below the Figure 6 that is "A", "B" and "C".
3.5 Relationships between characteristics of Resistant Starch
Chapter 4 is missing in this manuscript!
5. Conclusions - It is not really correct to place references to the researches of other scientists in the conclusions. So I would suggest rewording that sentence.

Author Response
To the Editors,
Thank you for the opportunity to revise and resubmit our manuscript entitled, Variation and abundance of Resistant Starch in Selected Banana Cultivars in Uganda. We appreciated the helpful comments from the editors and peer reviewers. Below, please find our itemized responses to the comments to accompany the revised manuscript we are submitting.
Thank you for your further consideration of this paper for potential publication in MDPI-Foods.
Sincerely,
Ali Kajubi
Reviewer 3 Comments to the authors:
I would recommend to supplement with the exact geographical coordinates of the banana growing regions. And in addition, this information is presented in the form of a table.
Response: The coordinates and altitude have been added as 1210m, 0o 24’ N and 32o 31’ E, Line 122 and the description of the analysed cultivars tabulated.
I would recommend revising the layout of the manuscript (eg the layout of chapter and subsection headings).
Response: The article layout has been revised using the journal template.
All equipment used must specify make, manufacturer, country. Check all equipment!
Response: The description of equipment used has been revised.
Did the experiment take place at exactly 40°C? Maybe there was ± some degree? For example, 40±1°C? Check all temperature!
Response: Whereas there are fluctuations in the temperature within the oven, it was set at exactly dried at 40°C for the drying of the starch and at 50°C for the isolated RS.
There are questions about the storage of this powder - in what package was it stored, where was it stored, under what conditions was it stored, how long was it stored?
Response: The resultant starch was stored at room temperature for not longer than a month. This has been added at Line 145-146.
How many repetitions were performed during the experiments? At what threshold were significant or non-significant differences found? "Principal component analysis" and “Cluster analysis” is also found in the manuscript, so this should be mentioned and explained in this section.
Response: All experiments were carried out in triplicate. Because the threshold of significance was different for different experiments, this significance level has been discussed in the individual results of the experiments. The PCA and cluster analysis have been mentioned as multivariate analysis in Line 240.
Is this the correct notation "cm−1"?
Response: The notation for wavenumber was revised to cm−1
The information I suggested should be given in the form of a table. It is very difficult to grasp now.
Response: The granule size data has been tabulated in Table 2, Line 253
Where can I see "Figure 2-2."? Only "Figure 2 A 2" and "Figure 2 B 2" are found in this manuscript.
Response: The description of the figure has been edited to Figure 2 A2 and B2, Line 315.
Maybe it's better to say "according to the Hoover study".
Response: Has been rephrased, Line 393.
I would recommend adding a reference, as the authors in this study have hardly investigated and proven what type of links and between which elements are formed or broken.
Response: Reference has been added as Abdullah et al., (2018) and Kumar et al., (2021), Line 411 and 412. In addition, the results from band assignment was added as supplementary data
I would suggest adding an explanation below the Figure 6 that is "A", "B" and "C"
Response: The description of A, B and C have been updated below the figure 6. Line 461
It is not really correct to place references to the researches of other scientists in the conclusions. So I would suggest rewording that sentence.
Response: The sentence has been reworded to ‘RS from Ndiizi and Kibuzi would be excellent for alternative application requiring transparent starches such as juice thickening agents.’ Line 517-519.

Round 2
Reviewer 1 Report
Comments and Suggestions for Authors
Accept.
Reviewer 2 Report
Comments and Suggestions for Authors
I am satisfied with this revision and hope that future studies will explore factors beyond species that influence RS from bananas.